# Valuation of the Diversity of Native Plants and the Cultural-Archaeological Richness as an Integrative Approach for a Potential Use in Ecotourism in the Inter-Andean Valley of Cusco, Southern Peru

**Isau Huamantupa-Chuquimaco** [1,*] , **Yohny Luz Martinez Trujillo** [2] **and Edilberto Orosco Ucamayta** [3]

1   Herbario Alwyn Gentry (HAG), Universidad Nacional Amazónica de Madre de Dios (UNAMAD), AV. Jorge Chávez N°1160, Puerto Maldonado 17001, Madre de Dios, Peru
2   Instituto Científico UAC, Programa ICDS, Universidad Andina del Cusco (UAC), Urb. Ingeniería Larapa Grande A-7, San Jerónimo, Cusco 17001, Cusco Region, Peru
3   Instituto Científico UAC, Universidad Andina del Cusco (UAC), Urb. Ingeniería Larapa Grande A-7, San Jerónimo, Cusco 17001, Cusco Region, Peru
*   Correspondence: andeanwayna@gmail.com

**Abstract:** In recent years, ecological tourism has become very important as it contributes significantly to sustainable development. In order to assess the potential for ecotourism and cultural-archaeological attributes, we studied the plant diversity of 10 traditionally visited natural routes of the valley of Cusco, Peru. Plant gamma diversity was represented by 384 species of vascular plants, with 220 genera, and 69 families; the most diverse were: Asteraceae with 93 species, Poaceae (36), and Fabaceae (15). The species with the highest frequency in the 10 routes are: *Amaranthus caudatus*, *Escallonia resinosa*, *Stenomesson pearcei*, and *Baccharis buxifolia*. Route 2 (Picol-Huaqoto) was the one with the greatest alpha diversity with 120 species. The CHAO-1 richness estimator estimates a gamma diversity of 570 species for all of the Cusco Valley. The Bray–Curtis beta diversity shows a high similarity (55%) and three floristic groups as determined by a non-metric multidimensional analysis (NMDS) and cluster analysis. The highest concentration of flowering plant species is grouped mainly during the rainy season ($R^2 = 0.19$), and this relationship is significantly different from the dry season ($p < 0.005$). The integrative biological–cultural analysis identified routes R8, R10, R6, R7, and R2 as those with the greatest potential for ecotourism use in the Cusco Valley. The plant diversity and cultural–archaeological offerings along the tourist routes documented in this study present significant opportunities for the city of Cusco to attract both national and foreign tourists. Additionally, this study highlights the importance of prioritizing conservation and preservation efforts for these areas.

**Keywords:** alpha diversity; Andes; tourist circuit; beta diversity; native plants; phenology; species richness

## 1. Introduction

Peru has world-renowned centers of high diversity such as the tropical Andes [1]. A high species richness of vascular and non-vascular plants was recorded in this region with southern Peru being a sub-region of the tropical Andes [2]. Pre-Columbian civilizations settled in this area, and over time Andean–Amazonian populations developed the use and management of many wild plants, placing the area among the most diverse in the world, similar to China, for example [3].

In the southern Peruvian region and in the Cusco Valley, located in the Huatanay River Basin, various pre-Inca cultures developed, mainly with the Inca culture best represented. The Inca culture, in the process of developing its geopolitical and cultural hegemony, developed extraction and deforestation activities, especially in lands destined for agriculture, as the basis of its expansion. This began with the establishment of the Marcavalle culture, approximately 800 years BC [4]. Socio-cultural development in the Cusco Valley to date

has involved various human activities that have displaced natural systems that have been transformed into crop fields, land for livestock, and urban and industrial areas [5]. This resulted in changes in land use, water pollution, soil degradation, habitat fragmentation, and the overexploitation of species, among other developments [6]. Evidently, the historic growth of cities in southern Peru has been predominantly informal and has therefore occurred in a disorderly fashion [7]. An example of this is the current population size of the city of Cusco, which currently has more than 450,000 inhabitants distributed mainly in the districts of Wanchaq, San Sebastián, and Santiago, few of which have adequate urban planning [8].

Regarding plant resources, the loss and massive degradation dates back mainly to colonial times with the overexploitation of tree and shrub species for use as fuel (firewood), construction, the expansion of agricultural and livestock areas, and the introduction of other exotic species, for example eucalyptus. Concerning this situation, some chroniclers, such as Garcilaso 1971 [9] and Valverde 1539 [10], suggest that when the Europeans arrived, the Cusco Valley corresponded to a great stony city with a warm climate surrounded by forests with abundant trees crossed by steamy ravines and rivers, where you could also recognize various forest tree species such as "mulli", "quishuar", "lambran", "pisonay", and "siwis", useful for housing structures and as fuel [11–13].

In recent years, tourism in the Cusco region has been one of the most important activities, which is directly linked to the extraordinary cultural and natural legacy of the Inca and pre-Inca cultures [14]. Furthermore, as an economic activity, ecotourism is one of the activities with less anthropic impact on the Cusco landscape, and has been prioritized worldwide [15]. Tourism has also generated benefits for local populations, as has been recognized in other areas [16].

Recently, in the province of Cusco and its surroundings, the use of biological resources in tourism has had a great growth; proof of this is that the different circuits or tourist routes mainly include cultural resources represented by structures and archaeological complexes such as temples, ruins, cobblestone roads, and housing constructions among others and nature attractions that include flora, fauna, ecosystems, and landscapes. In the valley of Cusco, five routes are recognized among the most frequently used routes for all visitors who travel in the city and valley of Cusco: the Sacsayhuaman Archaeological Park, Qenqo, Huanacaure, Tambomachay Hill, and Inkilltambo [17].

Despite the fact that some routes are traditionally recognized as the most frequently used, there are others that are also being implemented as new alternatives to the growing tourist offering to both locals and foreign visitors. For this reason, it is necessary to know what these routes are and the attributes that each one offers, with an integral criterion, which includes the biological and cultural components, since these are the most decisive when considering ecotourism activities.

The main objective of this study was to identify the potential of ten traditional natural routes for use in ecotourism activities in the inter-Andean valley of Cusco, for which the values of the biodiversity level (diversity of native plants and phenology patterns), anthropic impact, level of accessibility, and cultural components (number of archaeological remains) were used for their assessment.

## 2. Materials and Methods

### 2.1. Study Area

The Cusco Valley is located in the Huatanay river basin, within the Choco micro-basin and running for an approximate extension of 42 km to the confluence with the Vilcanota river (Figure 1). Throughout this area, the high Andean zone with woody vegetation is found in a range from 3200 to 4800 m above sea level. According to Marin 1961 [18], this area belongs to the Andean domain and the phytogeographic provinces of "Puneña" and "Mesoandina". Aragon and Chuspe 2018 [19] describe it as the ecoregion of Peruvian inter-Andean valleys, within the vegetation types of lowland forests and highland shrublands of the humid Puna and highland grasslands and scrub of the humid Puna (Figure 2).

Phytosociologically, we can describe the evaluated zones as being represented by the grassland in the upper zone, in the middle zone as a transition between shrubland and grassland, and the lower zones as dominated by trees, mainly by *Escallonia resinosa*. The climate is represented by two well-marked periods known as the dry season that goes from June to October and the rainy season from November to May; the average annual temperature is 10–13 °C and the area has an average total annual rainfall of 574–690 mm (average 12 years 1998–2020, Kayra weather station) [18,20,21].

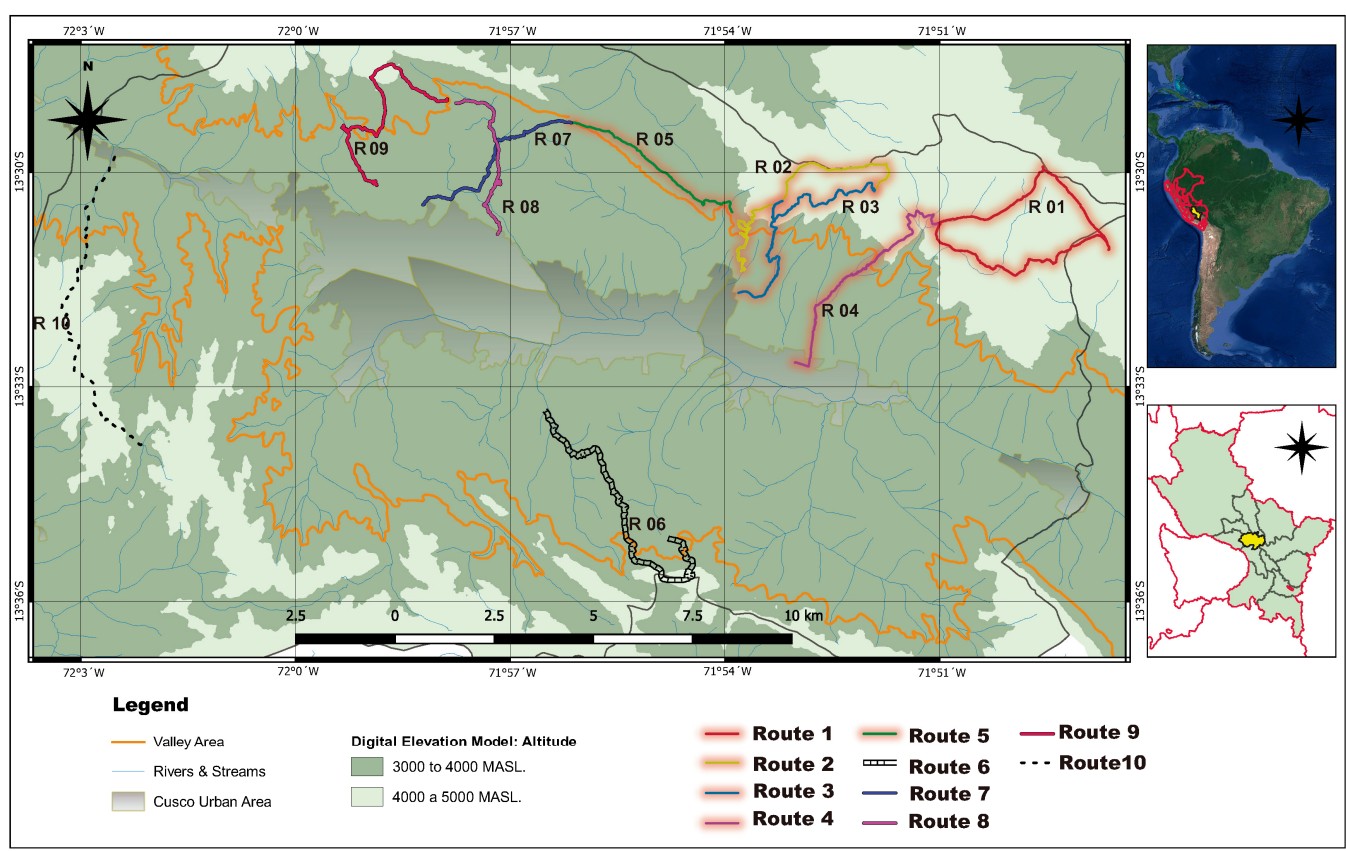

**Figure 1.** Distribution map of the touristic routes evaluated in the valley of Cusco.

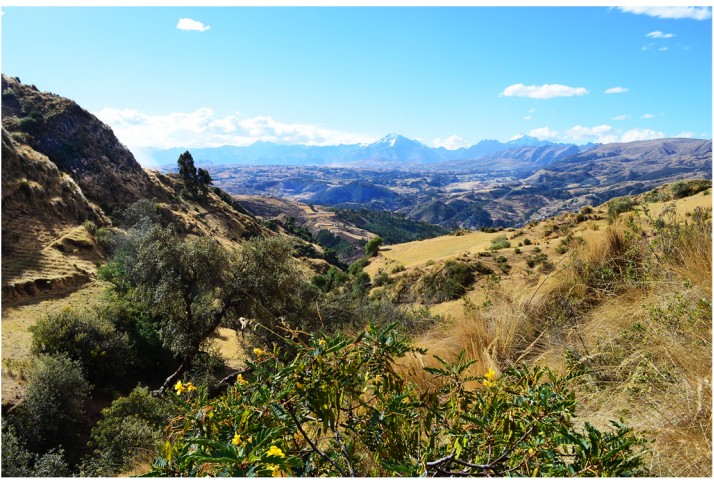

**Figure 2.** Characteristic high Andean landscape in the valley of Cusco, Peasant Community of Quishuarcancha—Cerro Mámá Simona—Poroy (Route 10).

## 2.2. Methodology

### 2.2.1. Selection and Evaluation of Ecotourism Routes

For our study, 10 routes (eco-trails), primarily those that have traditionally been the most frequently traveled by local and foreign visitors, were selected. This traditional selection took into account the following associated criteria over time: frequency of visits by local tourism (residents of the Cusco region), Peruvian nationals, and foreign tourists; observable biological diversity (plants and animals); number of cultural remains (pre-Inca and Inca archaeological complexes), and accessibility.

This study was carried out from October 2019 to July 2021. The 10 studied Routes were: Route 1 (R1) Huaqoto-Canteras de Huaqoto—Relicts of *Polylepis*—Path to Huanca—Huaqoto, Route 2 (R2): Tambillo—Cruz cerro Picol—Bosque de Santa María, Route 3 (R3): Tambillo—Cerro Picol—Picol Fault—Campos de San Jerónimo, Route 4 (R4): Huaqoto—San Jerónimo (along the pilgrimage road to Huanca, Route 5 (R5): Lomo de Iguana—Abra de Ccorao, Route 6 (R6): Rural Community of Kircas—Cerro Huanacaure—Inka Trail, Route 7 (R7): Yuncaypata—Temple of the Moon—Qenqo, Route 8 (R8): Pukapukara—Inkilltambo—Huayraqpunku, Route 9 (R9): Tambomachay—Balcón del Diablo—Sacsayhuaman, and Route 10 (R10): Quishuarcancha Peasant Community—Mama Simona Hill—Poroy. Each route had an extension of between 5370 and 12,700 m (Table 1). The altitudinal ranges of the routes varied from the valley floor between 3300–3400 to the maximum limits of 4500–4600 masl (Table 1, Figure 1).

**Table 1.** Quantitative and qualitative diversity data of native plants and the Cultural-Archaeological richness, evaluated in the 10 routes of the valley of Cusco.

| Routes | Trail Length (km) | Average Altitude | N° Archeological Units | Level of Anthropic Impact | Accessibility Level | Qualitative Richness (Observed) | Quantitative Richness (Sampled) | Individuals | Fisher's Alpha Diversity | Diversity q = 0 (Hill) |
|---|---|---|---|---|---|---|---|---|---|---|
| R1 | 12.70 | 4186 | 0 | 2 | 2 | 94 | 51 | 368 | 16.07 | 86 |
| R2 | 8.70 | 4081 | 0 | 2 | 2 | 120 | 82 | 326 | 35.23 | 110 |
| R3 | 8.94 | 3938 | 0 | 2 | 2 | 79 | 41 | 344 | 12.13 | 78 |
| R4 | 7.560 | 3702 | 1 | 2 | 1 | 97 | 50 | 322 | 16.57 | 80 |
| R5 | 5.420 | 3869 | 0 | 3 | 3 | 94 | 42 | 322 | 12.9 | 77 |
| R6 | 9.470 | 3719 | 2 | 2 | 1 | 108 | 64 | 200 | 32.55 | 86 |
| R7 | 5.390 | 3656 | 2 | 2 | 2 | 103 | 66 | 179 | 37.77 | 95 |
| R8 | 5.370 | 3594 | 2 | 1 | 1 | 111 | 70 | 130 | 61.82 | 124 |
| R9 | 7.690 | 3833 | 1 | 2 | 2 | 89 | 40 | 310 | 12.23 | 77 |
| R10 | 10.430 | 3947 | 2 | 1 | 1 | 109 | 71 | 264 | 31.86 | 108 |

Note: level of anthropic impact (High = 3, Medium = 2, and Low = 1); level of accessibility (Good = 1, Regular = 2, and Bad = 3).

### 2.2.2. Plant Diversity, Composition, and Phenology

In order to know the total richness of plants in each Route and the total existing in the 10 routes, we evaluated each one (on average 50 m from both sides of the route) of them by means of annotated lists, taking photographs, and, in the cases in which there were doubts about the identifications, we proceeded to collect them for later identification. Live form data (tree, shrub, and grass) and phenology (flowering and fruiting) as well as its vernacular name were recorded.

To estimate the richness, composition, and alpha and beta diversity, quantitative abundance data were collected; for this purpose, three modified Whittaker plots Campbell, 2002 [22] were installed on each route, each one located in the lower, middle, and upper zone of each route. We evaluated the trees in plots of $20 \times 50$ m (0.1 hectare), the shrubs in $5 \times 10$ m (50 m$^2$), and the herbs in 5 quadrats of $0.5 \times 2$ m$^2$ (5 m$^2$ total). The variables evaluated were the number of individuals (adults and developed juveniles), habit, crown area (coverage), phenology, and vernacular name.

For the taxonomic treatment of plant families, we used the reference established by the Angiosperm Phylogeny Group [23], as well as the herbaria CUZ, herbariums, and online digital repositories such as Tropicos (http://www.tropicos.org, accessed on 12 march 2020). For the taxa identifications, taxonomic keys and specialized bibliography were used, such as the Catalog of Angiosperms and Gymnosperms plants of Peru [24].

Alpha Diversity

The alpha diversity was calculated by two indicators: (a) the richness of each route was obtained from the qualitative data, representing the total number of species inventoried in each one, (b) the quantitative data of the total values of the three plots were used to calculate the Fisher's alpha index ($\alpha$). Fisher's alpha index is an abundance model derived from a logarithmic series that uses only the number of species (S) and the total number of individuals (N) [25]. This index is expressed as $S = \alpha \ln[1 + (N/\alpha)]$, where: S = number of species in the sample and N = number of individuals in the sample.

Furthermore, to estimate the total number of species that can be found on each route, we developed abundance curves using the rarefaction method, with the extrapolation function at the level of each route and subsequently for all routes. For this part of the analysis, we used Hill numbers q = 0 to quantify the effective number of all species, including rare species [26]. All these analyses were executed using the iNEXT package developed for the R software [27].

Beta Diversity and Similarity

The percentage of floristic similarity between plots was calculated using the quantitative Bray–Curtis index (obtained from abundance data) [28]. To visually represent these similarities, we used ordination methods and cluster analysis. The cluster analysis used the similarities and the group averages as a linkage method between groups and pairs of groups to build dendrograms. To complement this analysis, we further used a non-metric multidimensional scaling ordination (NMDS) [29]. The statistical analysis was performed in the program PAST v4.11 [30].

Gamma Diversity

To know the richness in the entire valley of Cusco, we summed up the total number of species inventoried in the 10 routes, and to estimate a projected total from the inventoried routes, the CHAO-1 non-parametric estimator was used (considering each of the 10 routes as units). This estimator prioritizes the presence of rare species within a sample (in this case each subplot), with the formula $\text{Chao1} = \text{Sobs} + ((n - 1/n)\,F1(F1 - 1)/2(F2 + 1))$, where: Sobs is the number of species observed in the whole valley, n the number of samples, F1 is the number of species observed in a single route (singleton species), and F2 is the number of species observed in two routes (doubleton species) [31,32]. The statistical analyses and graphs were made using the R software [27].

Phenology Patterns

To better understand the periods of greatest and least diversity, we have analyzed the phenological flowering patterns of all the plants on the 10 routes, which are also taken into account by visitors. For this, the differences between flowering individuals of each family, in the rainy seasons and those in the dry seasons, were evaluated with the non-parametric statistical comparison test of two independent Mann–Whitney samples. This analysis was expressed graphically in a box plot. To investigate if there was any relationship with environmental variables obtained from Kayra's meteorological station, located in the valley of Cusco, we carried out a linear regression analysis using the Spearman's non-parametric correlation coefficient [33] using the following dataset: the total species flowering each month, the data of total monthly precipitation, and the average monthly temperatures. The statistical analysis was performed in the program PAST v4.11 [30].

2.2.3. Comprehensive Consideration of the Potential of the Routes for Ecotourism

To obtain a weighting that helps us to identify the potential of each route for ecotourism activity in the Cusco Valley, for the present study we have considered the biodiversity component, mainly from the flora component, accompanied by ecosystem and cultural components. For this study, we have adapted the methodology known as multicriteria evaluation (EMC), frequently used in works related to the identification of biological,

cultural, and other potentialities for tourist resources. These criteria are described as "those that at the time of evaluation provide a logical and coherent way the contribution of the suitability criteria" for a potential proposal in ecotourism [34,35].

In order to know the routes with the greatest potential for use in the ecotourism activity in the Cusco Valley, we have considered four important criteria in each route: (a) the total richness of registered vascular plant species, which corresponds to the value of total richness; (b) the number of units of cultural remains; these refer to the presence of archaeological remains corresponding mainly to archaeological complexes including temples, ruins, rooms, and others belonging to the pre-Inca or Inca cultures and all these complexes are evaluated from the each routes; (c) the level of anthropic impact that the route has, which is qualified as low with the value of 1, regular with 2, and high with 3, which translates into whether the route mainly maintains the proportion in each route of the native vegetation and flora and the presence or absence of anthropic activities such as grazing, crops, or deforested areas without vegetation cover; and (d) the accessibility of the routes based on the fact that these routes have conditions for walking, such as appropriate distance and travel time; good accessibility is measured as having a value of 1, regular with 2, and not very adequate with 3 (Table 1).

## 3. Results

### 3.1. Diversity and Phenology

3.1.1. Richness, Composition, and Diversity

In the 10 routes, 384 species (qualitative richness) of vascular plants were found, distributed in 220 genera and 69 families, and 95% were identified at the species level and 5% at the morphospecies level. Regarding the habits, 17 species correspond to trees, 54 to shrubs, and 313 to herbs. The five Routes with the greatest floristic richness that exceed 100 species were R2 with 120 species, R8 (111), R10 (109), R6 (108), and R7 with 103 (Figure 1, Table S1). The families with the highest species richness were Asteraceae with 93 species, Poaceae (36), Fabaceae (15), and Rosaceae with 14 (Figure 3). The most diverse genera in terms of species were *Baccharis* with ten species, *Senecio* (nine), *Deyeuxia*, and *Plantago* with eight each. The species with the highest frequency of presence have been *Amaranthus caudatus* L. and *Escallonia resinosa* (Ruiz and Pav.) Pers., present in ten routes; present in nine routes were *Stenomesson pearcei* Baker, *Baccharis buxifolia* (Lam.) Pers., *Baccharis latifolia* (Ruiz and Pav.) Pers., *Baccharis odorata* Kunth, *Mutisia acuminata* Ruiz and Pav. (Figure 3D), *Puya ferruginea* (Ruiz and Pav.) L.B. Sm., and *Polylepis incana* (Kunth); and present in eight routes were *Amaranthus hybridus* L., *Chenopodium ambrosioides* L., *Chenopodium murale* L., *Grindelia boliviana* Rusby, and *Calceolaria myriophylla* (Kraenzl.) (Table S1).

In the quantitative sampling, a total of 2007 individuals were evaluated, comprising 252 species (quantitative richness) belonging to 54 families, the most diverse being Asteraceae with 67, followed by Poaceae (21), and Fabaceae with 12. The most abundant species were *Anatherostipa obtusa* with 173 individuals, *Lachemilla pinnata* (84), *Aciachne pulvinata* (71), *Lachemilla orbiculate* (58), and *Anatherostipa aff. obtusa* and *Muhlenbergia peruviana*, both with 57 individuals.

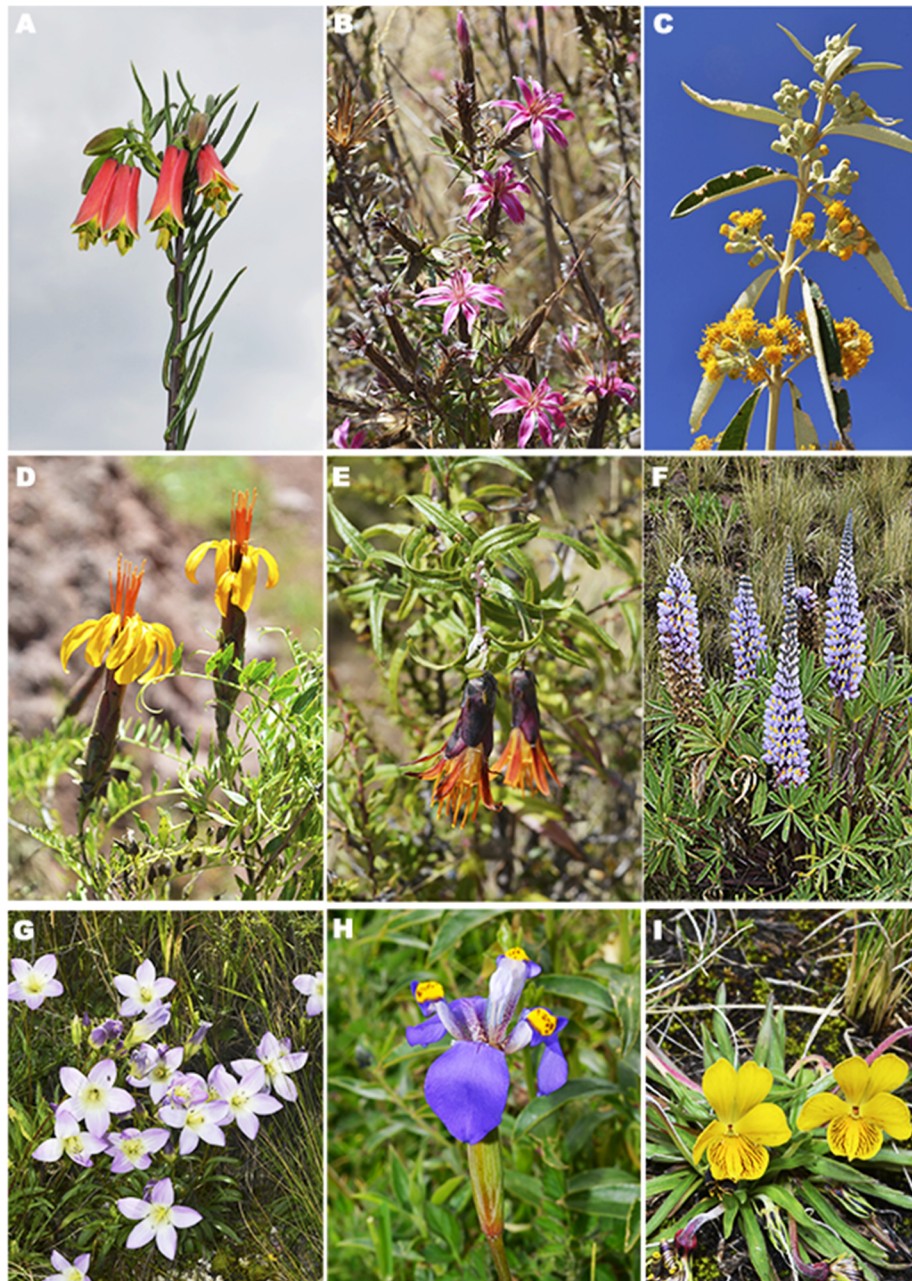

**Figure 3.** Species of native plants evaluated in the ecotourism routes in the Cusco Valley. (**A**) *Bomarea dulcis* (Alstroemeriaceae), (**B**) *Barnadesia horrida* (Asteraceae), (**C**) *Gynoxys longifolia*, (**D**) *Mutisia acuminata*, (**E**) *Mutisia cochabambensis*, (**F**) *Lupinus cuzcensis* (Fabaceae), (**G**) *Gentianella ernestii* (Gentianaceae), (**H**) *Hesperoxiphion herrerae* (Iridaceae), (**I**) *Viola aff. pygmaea* (Violaceae).

Alpha Diversity

The route with the greatest diversity of vascular plants was R8, with Fisher's alpha = 61.82, followed by R7 (Fisher's alpha = 37.77), R2 (Fisher's alpha = 35.23), R6 (Fisher's alpha = 32.55), and R10 (Fisher's alpha = 31.86). The other routes obtained values less than 30 Fisher's alpha (Table 1). The estimated richness for each route, through Hill's diversity (q = 0), indicates that the Route with the greatest diversity was R8 with 124 species, followed by R2 with 110 and R10 with 108 species; for the other routes the richness was estimated to be less than 100 species (Table 1, Figure 4).

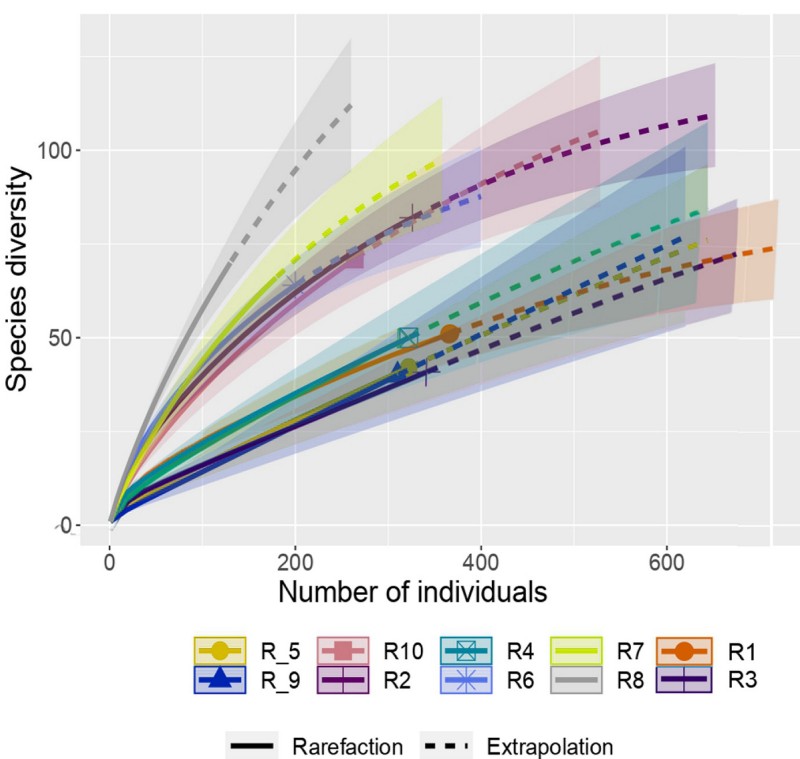

**Figure 4.** Diversity using the rarefaction method based on the Hill index.

Beta Diversity and Floristic Affinities

The cluster and the NMDS plots show the formation of three main groups of routes. In the cluster with 55% similarity (Figure 5a,b), group 1 is made up of routes R2, R4, R6, R7, R8, R9, and R10; group 2 by R1 and R3; and group 3 is represented only by R5. This same pattern is evident in the NMDS with the three groups with low stress (0.13), which indicates a good fit of the model to the data (Figure 5b). In the first dimension, group 1 is concentrated in the central part, with the most distant group being 2 with R1 and R3, and finally, at the end of the axes, there is only R5 as group 3.

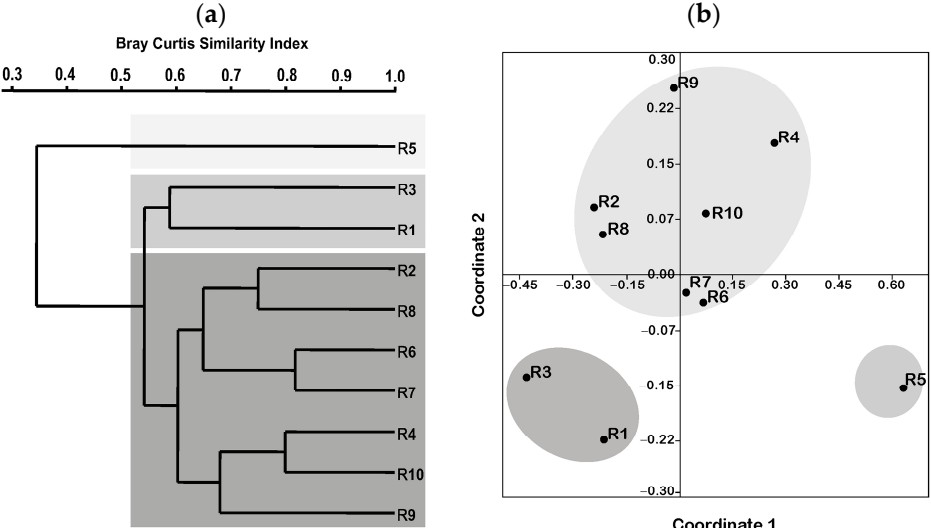

**Figure 5.** (**a**) Similarity between the 10 sampled routes in the Cusco Valley. (**b**) NMDS showing the grouping of the floristic groups between 10 sampled routes in the Cusco Valley.

Gamma Diversity

The total diversity observed for the Cusco Valley, from the 10 routes, is represented by 384 species. The quantitative data, through the rarefaction curve by means of the CHAO-1 index, estimates that for the entire Cusco Valley there is a richness of 570 species of vascular plants (Figure 6).

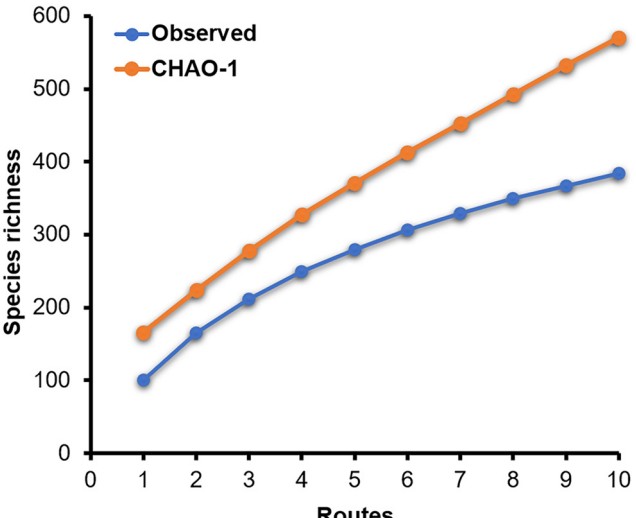

**Figure 6.** A. CHAO-1 diversity, observed and estimated.

3.1.2. Phenology Patterns

Phenology patterns show the largest number of species are flowering during the months of the rainy season, from February to July. Then, there is a gradual decrease in the dry season, where only some species flower. Few biennials have more than one flowering season within the year (Table 2). Herbs mainly flower from December to May and most of the shrubs also concentrate their flowering during these months. In the case of trees, it is observed that there is a continuous flowering with the exception of June and August, where no species was recorded to flower (Table 2, Figure 7).

**Table 2.** Number of flowering species per habit and per month during the year 2020. ND = no data available.

| Habit | January | February | March | April | May | June | July | August | September | October | November | December |
|-------|---------|----------|-------|-------|-----|------|------|--------|-----------|---------|----------|----------|
| Tree | 8 | 12 | 8 | 10 | 12 | ND | 2 | ND | 2 | 2 | 2 | 5 |
| Shrub | 19 | 41 | 37 | 36 | 40 | 2 | 4 | 3 | ND | 7 | 7 | 15 |
| Herb | 103 | 203 | 206 | 203 | 199 | 26 | 37 | 15 | 13 | 21 | 37 | 107 |
| Total | 130 | 256 | 251 | 249 | 251 | 28 | 43 | 18 | 15 | 30 | 46 | 127 |

The months with the greatest diversity of flowering plants are February (256 species), March and May (251), and April (249). The months in which there are fewer species are August and September (Table 2, Figure 6).

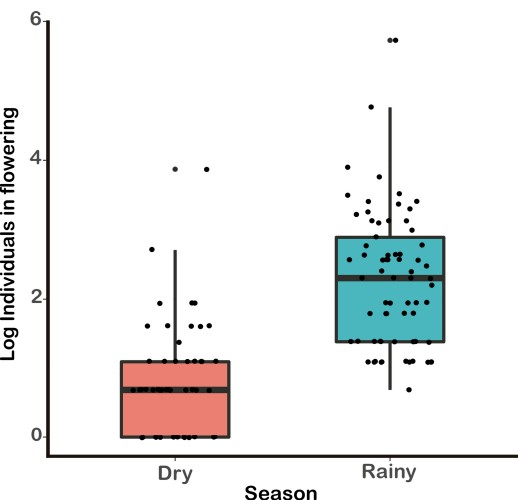

**Figure 7.** Phenological variation between the two seasons (Mann–Whitney test *p* = 0.00001).

Throughout the year, the botanical families that contributed the most flowering sightings during the year were Asteraceae with 307 observations, Poaceae with 117, Fabaceae 49, Rosaceae 43, and Plantaginaceae 34 (Table 3).

**Table 3.** List of the 10 most representative families with individuals in flowering, in the dry and rainy seasons (the values correspond to the sums of observations in the dry and rainy months).

| Family | Rainy Season | Dry Season |
|---|---|---|
| Asteraceae | 307 | 48 |
| Poaceae | 117 | 15 |
| Fabaceae | 49 | 5 |
| Rosaceae | 43 | 7 |
| Plantaginaceae | 34 | 7 |
| Orobanchaceae | 33 | 7 |
| Amaranthaceae | 30 | 5 |
| Amaryllidaceae | 30 | 3 |
| Gentianaceae | 29 | 5 |
| Orchidaceae | 27 | 1 |

For the total 69 families, in the rainy season 1264 flowering individuals (mean 18.31; standard deviation 38.84) were observed, and for the dry season 180 individuals were observed (mean 2.60; standard deviation 6.05). The Mann–Whitney test shows that there is a significant difference between the number of observations of individuals in flower in the rainy and dry seasons (*p* = 0.00001). Therefore, the rainy season is more favorable to appreciate a high richness of species and individuals in the evaluated routes (Figure 7).

Regression analysis shows us that there is an association between richness and precipitation in the evaluated 12 months, with a $R^2$ = 0.19 Spearman's correlation coefficient of *p* = 0.15, which is considered a low to regular association (Figure 7a).

The relationship between species richness and temperature shows a $R^2$ = 0.02 Spearman's correlation coefficient of *p* = 0.48, which is considered a low association (Figure 8a,b). Between temperature and precipitation during the studied 12 months, we observed that precipitation is the best variable that explains the increase in plant richness in the rainy months of the year.

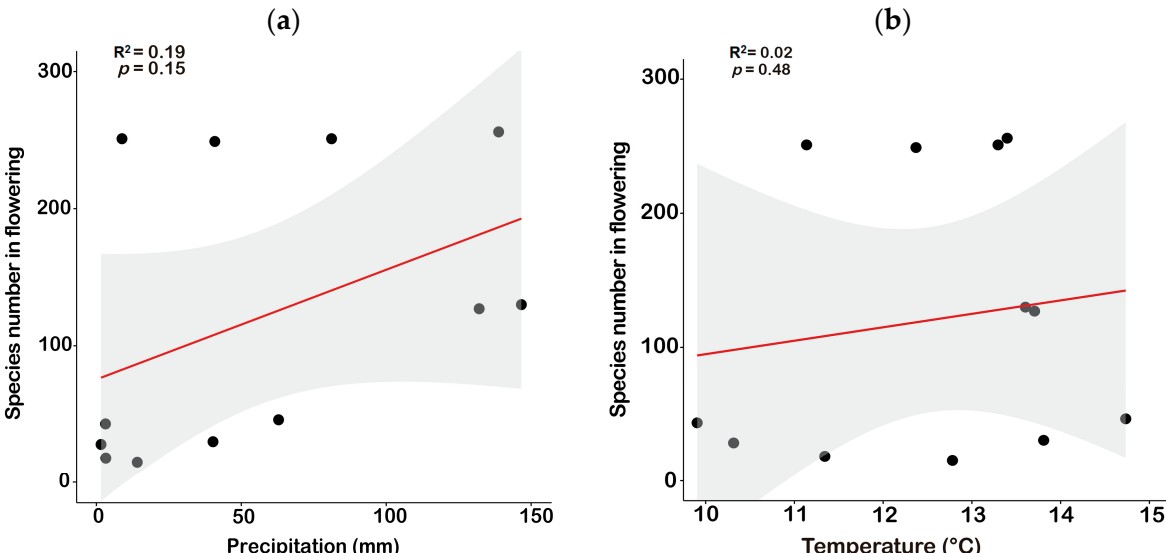

**Figure 8.** Representation of the annual relationship between environmental variables and the richness. (**a**) Precipitation vs. species richness, (**b**) Temperature vs. species richness.

### *3.2. Potential of Routes for Ecotourism*

In the 10 routes, 10 archaeological units were registered. In R4, the Raqayraqayniyuq complex is present (San Jerónimo district); R6 has two units: the Huanacaure hill complex and the Inca trail; R7 has the archaeological complexes of the Temple of the Moon (Amarumarkawasi) and Qenqo; R8 includes the Puka Pukara and Inkiltambo complexes; R10 has the Inca trail to Apu Mama Simona and the Quiswarcancha complex; R4 has a single complex Inca trail to Huacoto; and R9 has part of the Sacsaywaman archeological park.

The five routes with the highest weighting considering the four criteria are Route 8 (total richness = 111 species, archaeological units = 2, low anthropic impact = 1, and with good accessibility = 1); Route 10 (total richness = 109, archaeological units = 2, anthropic impact = 1, and with good accessibility = 1); Route 6 (total richness = 108, archaeological units = 2, anthropic impact = 2, and with good accessibility = 1); Route 7 (total richness = 103, archaeological units = 2, anthropic impact = 2, and with regular accessibility = 2); and Route 2 which, despite not having any archaeological unit, presents the greatest diversity of all the routes with 120 species, regular anthropic impact (2), and regular accessibility (2) (Table 1).

The other routes, with the exception of R4 and R9, do not present any archaeological units. However, in general, they have access conditions, and are regularly impacted by human activities, mainly agricultural activity.

### 4. Discussion

### *4.1. Diversity and Phenology*

#### 4.1.1. Richness, Composition, and Diversity

Gamma diversity is represented by 384 species of vascular plants found in the 10 routes of the valley of Cusco, and shows in general that these types of vegetation, that include the inter-Andean dry forest in the lower zone and the "pajonales" in the highlands, encompass quite representative habitats that include transition areas. This is evident in the lower areas with the presence of forests dominated by populations of native trees: *Escallonia resinosa* (chachacomo) and *Polylepis racemosa* (queuña), and in the grasslands dominated by the grass *Anatherosthipa obtusa* (ichu). These routes can each house up to more than 100 species. The total richness or gamma diversity of the plants registered in the present study is notably higher than that documented by Herrera 1941 [12], who points out that, at that time, in the surroundings of the city of Cusco there were up to 254 species of higher plants. This approximation somewhat calls attention to the fact that, in those years, the

city of Cusco still had many remnants of forest and more intact areas near the banks of the micro-watersheds throughout the entire valley. Compared with the study by Galiano et al., 2005 [36], who registered little more than 480 species for the entire Cusco Valley, in our study more than 80% of them are cataloged. It should be noted that the study of these authors, in addition to the Cusco Valley, included the towns of Urcos and Huacarpay, which are not directly part of the Cusco Valley.

Family diversity patterns, both in qualitative and quantitative evaluations, show similar results as found in other regions of the Cusco Valley, as described by the works of Galiano et al., 2005 [36]; Tupayachi 2019 [13]; Herrera 1941 [12]; and Marín 1950 [18]. In these studies, the families with the greatest diversity were Asteraceae, Poaceae, Fabaceae, and Lamiaceae, the exception being the Bromeliaceae family. In the present study, we only registered two species in the different routes. The same authors mention that, in general, for the forests and grasslands of the Cusco Valley and its surroundings, these genera are more representative: *Deyeuxia* (ex *Calamagrostis*) of the Poaceae, *Puya* (Bromeliaceae), *Solanum* (Solanaceae), *Senecio* (Asteraceae), and *Lupinus* (Fabaceae).

In the 10 studied routes, the species considered to be most frequent present interesting distributions, for example *Amaranthus caudatus*, *A. hybridus*, *Chenopodium ambrosioides*., and *C. murale* have a wide range of distribution and can be found even in degraded environments, which shows that along the routes these species were also adapting alongside wild species from habitats that were little disturbed. The other group of frequent species are trees, such as *Escallonia resinosa*, *Baccharis buxifolia*, *Barnadesia horrida* (Figure 3B), *B. latifolia*., *B. odorata*, *Mutisia acuminata* (Figure 3D), *Polylepis incana*, and the herbaceous species *Stenomesson pearcei*, *Grindelia boliviana*, and *Calceolaria myriophylla*, which are from little impacted habitats and which are also quite frequent in other inter-Andean valleys of southern Peru. At a quantitative level, in the 30 plots the pattern of abundance is different in terms of the individuals evaluated. The species *Anatherostipa obtusa*, *Lachemilla pinnata*, and *Aciachne pulvinata* stand out as these are species of greater density and small size. These species are present mainly in the highlands such as grasslands and transition zones within the inter-Andean dry forest.

The gamma diversity values estimated by the CHAO-1 index show that the richness of vascular plants can reach around 570 species for the Cusco Valley; this value is comparable to that recorded in Galiano et al., 2005 [36] with little more than 480 species in the vicinity of the Cusco Valley. At the level of each route, if we compare the values of Fisher Alpha with other Andean zones such as in the departments of Cajamarca and Pasco above 2800 masl, they present indices between 12 and 60 [25] which indicates that the Cusco routes in general contain a similar moderate to low diversity, which are also relatively similar to other Andean regions. The values found are also corroborated with the diversity values of Hill where three routes, R2, R8, and R10, estimate the presence of more than 100 species, which are complemented by the qualitative results, where routes R2, R10, R8, R6, and R7 exceed more than 100 species. These values and estimates show that although the ecotourism routes described are often not well recognized with high species richness, with more detailed sampling, areas with high richness and alpha diversity can be recorded, especially if we make a complete evaluation during a year.

The beta diversity evaluated with the UPGMA and NMDS methods, under the Bray–Curtis index, generally exhibits a high floristic similarity among the 10 routes, that is, more than 50% of species are shared among all of them. However, three consistent groups are distinguished, of which group 1, with 7 routes (R2, R4, R6, R7, R8, R9, and R10), represents those that include valley bottom areas, inter-Andean dry forest, and grassland areas in their routes, while group 2 (R1, R3) corresponds to drier areas with few tree populations in the lowlands, and group 3 is represented only by route R5 and contains less floristic richness since it only covers habitats such as grassland and shrubland with a greater anthropic impact.

4.1.2. Phenology

Apparently, the monthly flowering periods are conditioned by the presence of rain within a year, and it is within this period where more flowering species could be seen. This same pattern is mentioned by Marín 1950 [18], who made some notes on the floral phenology in different months of the year in the surroundings of Cusco of the arboreal, shrubby, and herbaceous flora, reaching the conclusion that the month of March has the largest floral anthesis in Cusco, which is also considered to be a widespread pattern in the Andean region of Peru.

Groups such as the Asteraceae, Poaceae, Fabaceae, and Rosaceae are the ones that show a greater flowering pattern in the rainy months. Species such as *Viguiera procumbens*, *Ageratina cuzcoensis*, *Begonia veitchii*, *Gentianella ernestii* (Figure 3F), the mostly perennial *Barnadesia horrida* (Figure 3C) and *Mutisia acuminata* (Figure 3D), and the endemic *Lupinus cuzcensis* (Figure 3G) are the ones that can be recorded in an abundant way during the flowering season in the areas mainly of the inter-Andean dry forest at the bottom of the valley. These are even used as festive decorations in carnivals, as a symbol of prosperity in the year.

*4.2. Integral Assessment of the Potential and Sustained Management of Routes for Ecotourism*

Sustainable tourism is one of the activities that has been providing opportunities to local populations [16]. Ecotourism in recent years has caused the least negative impacts on the environment, compared with other activities [15]. In the area of the Cusco Valley, tourist activity is represented by routes recognized as "Cusco City tour", "Cusco Valley tourism", and "Sacred Valley tourism". For Cusco, the most traveled routes are Inkiltambo, Qenqo, Sacsayhuamán Archaeological Park, Apu Huanacaure, and Apu Pikol [17].

In the present study, we considered the integral biological, ecosystemic, and cultural criteria, which support our conclusion that the routes R8, R10, R6, R7, and R2 are the ones with the greatest potential for use in ecotourism within the Cusco Valley. These routes also include the most important archeological complexes in the peri-urban area of the Cusco Valley, as indicated in [17]. These findings agree with the hypothesis that the Cusco Valley, with its micro-climatic variability, geological formations, ecosystems, and fertile lands, has been the reason the area was chosen for the capital city by the Incas and other pre-Inca cultures, despite being above 3000 masl, as stated in Galiano 2005 [11] and Marín 1950 [18].

The studied ecotourism routes, apart from the impressive diversity of plants (384 species and an estimated 570), were traditionally areas where resources such as construction material, medicinal plants, and foodstuffs, among others, were obtained in an important way in the daily life of the "cusqueño" inhabitant in practices which are currently still being practiced (field observation).

Despite the fact that these routes have a high potential for ecotourism use, it is important to highlight that some of them are subject to risks and threats, mainly due to the expansion of housing construction, the authorization of cultivation areas, and fires that occur mostly during dry months. To contribute to its conservation, we propose the following measures for the recovery and restoration of the populations of native forests and associated flora on the evaluated routes: (1) an area on the margins at each route that includes forests and remnants in which strict protections are respected; (2) potential restoration areas are identified to recover deforested areas that currently correspond to deteriorated areas such as in the lower areas of R2, R4, and R10; (3) the preparation and implementation of a plan to replace *Eucalyptus* plantations (*Eucalyptus globulus*) by native forest species, taking into account historical records, such as those species present in the R10, which includes highly threatened species such as *Gynoxys longifolia* (Figure 3C) and *Escallonia myrtilloides*, both native forest species that were decimated in the past in the vicinity of the city of Cusco; (4) the implementation of education campaigns, studies, and programs for the reevaluation of the native flora in the valley of Cusco; (5) the consideration of intangible conservation areas to avoid inadequate access by tourist visitors, for example, areas where endemic plants with extinction threats grow, nesting areas for some birds such

as partridges and hummingbirds with a degree of threat, and mammal feeding areas, for example, the taruka. Within these areas, we have identified the upper parts of the grassland, where you can find representative animal species with high degrees of threat, such as the taruka "*Hippocamelus antisensis*", the Andean fox "*Lycalopex culpaeus*", and the Andean partridge "*Nothoprocta ornata*".

## 5. Conclusions

In the present work, we show that in the Cusco Valley there is a high gamma diversity of vascular plants, with a projected 570 species, in 10 ecotourism routes traditionally traveled by local and foreign populations in the peri-urban areas of the city of Cusco. The beta diversity also shows a high plant similarity between the 10 routes, where a group of routes show a consistent grouping corresponding to those with the highest alpha diversity such as R2, R6, R7, R8, and R10. Due to the climatic characteristics of this part of the Andean region, the greatest diversity of plants in flowering season can be observed during the rainy season, which corresponds to the first months of the year.

The potential use of each evaluated route, by applying the integrative method, which in our case was based on biological (represented by the diversity of plants), ecosystem (with the level of anthropic impact, and its accessibility), and cultural values (with the number of cultural units present in each route, which correspond to legacies of the rich Inca and pre-Inca cultures), has allowed us to assign an appropriate weighting to each route. The routes with the highest values were R2, R6, R7, R8, and R10. These routes have a high diversity of plants, the presence of up to two cultural units, a low level of anthropic impact, and adequate accessibility. In general, for the 10 routes, it is recommended that some mitigation measures be implemented, such as the restoration of degraded environments of natural habitats, reforestation, ecological connectivity areas, and strict protection areas. For the sustainable use and management of these areas, local authorities must also consider the establishment of rest areas, signage, the preparation of local guides, and zoning. These areas undoubtedly represent a good recreation alternative and the opportunity for programs for environment education and an appreciation of nature for the population of the city of Cusco as well as national and foreign tourists.

**Supplementary Materials:** The following supporting information can be downloaded at: https://www.mdpi.com/article/10.3390/d15060760/s1, Table S1: General table of species with collections by routes.

**Author Contributions:** Research conception and design: I.H.-C. and Y.L.M.T.; field study and sampling: I.H.-C., Y.L.M.T. and E.O.U.; analysis and interpretation: I.H.-C. and E.O.U.; drafting of the manuscript: I.H.-C. All authors have read and agreed to the published version of the manuscript.

**Funding:** The present investigation was developed thanks to the financing obtained by the "Project for the Improvement and Expansion of the Services of the National System of Science, Technology and Technological Innovation" 8682-PE, from the "World Bank" to "CONCYTEC" and "PROCIENCIA" as financing entities of the project "From trekking routes to ecotourism routes/paths: adaptation of trails in Andean and high Andean areas for the recreation of the local population, Case: Cusco—Peru", which was financed by CONCYTEC–PROCIENCIA within the framework of the call E041-01 (N°148-2018 FONDECYT-BM-IADT-AV).

**Institutional Review Board Statement:** Not applicable.

**Data Availability Statement:** Not applicable.

**Acknowledgments:** To Di Yanira Bravo, for her support in managing the viability of the Ecotrails project through CONCYTEC and the research institute of the Andean University of Cusco. To Edwin Bellota, for his collaboration in the identification and selection of the 10 ecotourism routes in the Cusco Valley, and for the suggestions to the manuscript. To the members of the UNSAAC ECOTAXON Research Center: Anne Arias, for her collaboration and assistance in collecting data in the field, and Armando Cuti, Jean Pier Quispe, Rolando Chaparrea, Rodrigo Calvo, Hugo Copa, Fany Cutire, Jhon A, Yuca, Esau Huaman, Elias Quispe, Valentina Palomino, Flora Sucsa, Mario Sanchez, Nidia

Sánchez, and Williams Navarro, for their assistance in field sampling and logistical support. Thanks to Roosevelt García-Villacorta for providing suggestions for an early version of the manuscript. To the inhabitants of the communities immersed in the ecotourism routes of the districts of San Sebastián, San Jerónimo, and Ccorao of the province of Cusco.

**Conflicts of Interest:** The authors declare no conflict of interest.

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
