# Peer review of "Valuation of the Diversity of Native Plants and the Cultural-Archaeological Richness as an Integrative Approach for a Potential Use in Ecotourism in the Inter-Andean Valley of Cusco, Southern Peru"

_diversity, doi:10.3390/d15060760_

Round 1

Reviewer 1 Report

This manuscript presents a very interesting study on the potential of routes/trails for ecotourism in the Cusco-valley based on an assessment of plant diversity, cultural variables, human impact and accessibility and gives a final ranking of possible routes for eco-tourism development in the future.

I like the general approach of the manuscript and would support publication. There are however some inconsistencies with terms and definitions. I also miss a critical evaluation of ecotourism. Even though it is stated that the impact of this form of land use was least severe, it still has an effect that might increase with higher intensification of tourism. Therefore, it should also be taken into account that if species are present with a high category of threat, it could also be an option to restrict access due to conservational reasons.

In the following some specific comments:

L35-36: Please rephrase this sentence, e.g.: A high species richness of vascular and non-vascular plants was recorded in this region with southern Peru being a sub-region of the tropical Andes.

L51: [7].

L66-67: This sentence is difficult to understand. You mean that compared to other landuse forms tourism had the least impact? But on what? On the landscape? On natural ecosystems or on native diversity?

L99-102: For which time span the average climate data are given?

L125: I doubt that the routes are 5000 to 12000 km long.

L129-133: How wide was the area from the trails that was investigated for assessing the floristic diversity?

L137: 20 x 50 m

L138: 0.5 x 2 m

L147: Please differentiate more between the different scales that you regarded. You have determined the alpha diversity for each route, but also across routes it seems that you name everything alpha diversity. This is confusing. Summing up the species across routes results in gamma diversity of the valley.

L175: Was the NMDS ordination based on two axes? Was this also done with the R software? Which package and which function was used then?

L182: Delete which

L184-187: Please rephrase, e.g.: ….the valley of Cusco, we correlated the total number of species flowering in each month with monthly precipitation and the average monthly temperatures using the Spearman non-parametric correlation coefficient.

L186: A linear regression is not a correlation. When you used the Spearman coefficient then you did a correlation and no linear regression. However, in the results chapter you mention the Pearson correlation. What have you used?

Table 1: Trail length is not 12000 km! In this table you now use different terms for your diversity variables compared to the methods. This is really confusing. What is qualitative richness? Is this the observed richness? The term was not used before. To which spatial scale the term species richness refers to? Please adjust the nomenclature of spatial scales and diversity levels across the manuscript.

L271: Delete that

L276: The months with the greatest diversity of flowering plants are February (256 species), March(251)…

Table 2: The table heading is a bit misleading. It is rather the number of flowering species per month.

L294: In the methods you said Spearman correlation, here it is Pearson. What have you used? And is this really r² or is it the correlation coefficient r? The relationship in the first graph does not look like an r² of 0.43. And are the displayed relationships significant?

Figure 7: Please be more precise with naming the axis. Here Y is the "Number of flowering species per month across the routes" and x is the monthly precipitation or temperature.

L312-319: Maybe it would be good to present a ranking list also in table 1 by combining the different variables displayed in the table. Also you talk about weighting but it is not clear if there was a weighting at all. Have you given a certain aspect a higher importance, e.g. diversity?

L326: See my earlier comments. This is not alpha diversity. This should be described as gamma diversity as this is the total richness across the sampled routes. 

L347: Delete which and start a new sentence: In our case, we only registered…

L366: Here again the term alpha diversity even though it again refers to the total gamma diversity of the valley estimated with Chao estimator.

L426ff: I am missing a small paragraph that also discusses the potential risk of more tourists taking these routes. If for example, some very rare species occur along the routes, there is also a risk that they will be negatively affected by tourists and the question is, if the access to some routes with a high conservation value should maybe be restricted for conservational reasons. It just should be discussed somewhere that tourism is not only positive.

L449: Again alpha diversity. There is a strong confusion with terms. Please adjust the terms to the different spatial scales that you have observed.  

L459: cultural variables

L465: should be implemented

L472: I could not find the supplement with the submitted manuscript.

Author Response

Comments of revisor 1

L35-36: Please rephrase this sentence, e.g.: A high species richness of vascular and non-vascular plants was recorded in this region with southern Peru being a sub-region of the tropical Andes.

Was implemented

L51: [7].

Was implemented

L66-67: This sentence is difficult to understand. You mean that compared to other landuse forms tourism had the least impact? But on what? On the landscape? On natural ecosystems or on native diversity?
Was implemented

L99-102: For which time span the average climate data are given?

L125: I doubt that the routes are 5000 to 12000 km long.

Was implemented

L129-133: How wide was the area from the trails that was investigated for assessing the floristic diversity?

Was implemented

L137: 20 x 50 m

It was implemented

L138: 0.5 x 2 m

Was implemented

L147: Please differentiate more between the different scales that you regarded. You have determined the alpha diversity for each route, but also across routes it seems that you name everything alpha diversity. This is confusing. Summing up the species across routes results in gamma diversity of the valley.

Was implemented

L175: Was the NMDS ordination based on two axes? Was this also done with the R software? Which package and which function was used then?

Was clarified

L182: Delete which

L184-187: Please rephrase, e.g.: ….the valley of Cusco, we correlated the total number of species flowering in each month with monthly precipitation and the average monthly temperatures using the Spearman non-parametric correlation coefficient.

Was implemented

L186: A linear regression is not a correlation. When you used the Spearman coefficient then you did a correlation and no linear regression. However, in the results chapter you mention the Pearson correlation. What have you used?

Table 1: Trail length is not 12000 km! In this table you now use different terms for your diversity variables compared to the methods. This is really confusing. What is qualitative richness? Is this the observed richness? The term was not used before. To which spatial scale the term species richness refers to? Please adjust the nomenclature of spatial scales and diversity levels across the manuscript.

Was implemented

L271: Delete that

Was implemented

L276: The months with the greatest diversity of flowering plants are February (256 species), March(251)…

Was implemented

Table 2: The table heading is a bit misleading. It is rather the number of flowering species per month.

Was implemented

L294: In the methods you said Spearman correlation, here it is Pearson. What have you used? And is this really r² or is it the correlation coefficient r? The relationship in the first graph does not look like an r² of 0.43. And are the displayed relationships significant?

Was clarified

Figure 7: Please be more precise with naming the axis. Here Y is the "Number of flowering species per month across the routes" and x is the monthly precipitation or temperature.

Was clarified

L312-319: Maybe it would be good to present a ranking list also in table 1 by combining the different variables displayed in the table. Also you talk about weighting but it is not clear if there was a weighting at all. Have you given a certain aspect a higher importance, e.g. diversity?

Was clarified

L326: See my earlier comments. This is not alpha diversity. This should be described as gamma diversity as this is the total richness across the sampled routes. 

Was clarified

L347: Delete which and start a new sentence: In our case, we only registered…
Was implemented

L366: Here again the term alpha diversity even though it again refers to the total gamma diversity of the valley estimated with Chao estimator.

Was implemented

L426ff: I am missing a small paragraph that also discusses the potential risk of more tourists taking these routes. If for example, some very rare species occur along the routes, there is also a risk that they will be negatively affected by tourists and the question is, if the access to some routes with a high conservation value should maybe be restricted for conservational reasons. It just should be discussed somewhere that tourism is not only positive.

Was clarified

L449: Again alpha diversity. There is a strong confusion with terms. Please adjust the terms to the different spatial scales that you have observed.  

Was clarified

L459: cultural variables

Was clarified

L465: should be implemented

Was implemented

L472: I could not find the supplement with the submitted manuscript.

Was implemented

Reviewer 2 Report

The manuscript diversity-2215779 (“Valuation of the diversity of native plants and the cultural-archaeological richness as an integrative approach for a potential use in ecotourism, in the inter-Andean valley of Cusco, southern Perul”) where the authors studied the plant diversity of 10 traditionally visited natural routes of the valley of Cusco and concluded the opportunities for the city of Cusco to attract both national and foreign tourists.

The introduction is good. The references is very good and the history very well explained.

The sampling method should explain better. Regarding the methodology, the authors need to redo many things such as adding more plots as the number of plots they indicate is insufficient (only 3 per route). I do not understand why it is necessary to calculate phenology if it is not the objective of the work (that could be another independent work). With respect to the archaeological part, I don't understand how they have obtained this information and finally, the obtaining of qualitative variables for the anthropic impact is not very clear.

Results should improve. The authors should increase the number of plots on each route. The experimental design is poor. The pheonology is not related to the objective of the work. To answer the question, analysing the diversity and composition of the vegetation is sufficient. They should better detail how to obtain temperature and precipitation data (through interpolated maps, data logger,...). Regarding cultural aspects, I don't see any analysis in the results that talks about it.

The discussion is not bad but it should be improved when the new analyses are redone and the number of plots is increased, as 3 plots per route is not enough, especially when there is a lot of heterogeneity in the routes.

Some comments:

Line 97: What vegetation type in this region? Please, indicate the community’s vegetation (Phytosociology).

Line 100: The authors comment that two well-marked periods. Please, the authors should indicate the average annual temperature and annual rainfall in this two periods.

Figure 1: The author should include scale and orientation in the two small maps on the right side of figure.

Line 11: Are there any statistical surveys that indicate that these places are the most visited sites? On what basis? On direct observation?

Line 137: Change “20 x 50 mm” by “20 x 50m”. And it’s better change “0.1 hectare” by “1000 m2”.

Line 137: On such a long distance (e.g. 10-12 km of a route) the number of plots is insufficient. Only 3 plots you can't conclude anything concrete as you have too few sampling points. What is the distance between plots? Please clarify…

Line 139: I understand that “crown area” is cover (%), it’s correct? Please clarify…

Line 139: Numbers of individual included all individuals (juveniles and mature)?

Line 169: Other analysed that I recommend to the author is a DCA (detrended correspondence analysis) with CANOCO or R software.

Line 176:  I do not understand what the flowering period has to do with the objective of the work, which is to identify the potential of ten traditional nature trails to be used for ecotourism activities in the inter-Andean valley of Cusco. The authors are not answering the question with this section.

Line 185: How did you obtain data on precipitation and average temperatures? Have you placed dataloggers on the plots to obtain this information? Please clarify…

Lines 201-203: With respect to number of units of cultural remains, these refer to the presence of archaeological remains corresponding mainly to archaeological complexes including temples, ruins, rooms and others belonging to the pre-Inca or Inca cultures, the authors do not indicate where they have obtained this information. Please clarify…

Line 203: How has the anthropogenic impact of each route been studied? What variables have the authors used? Have they not taken into account quantitative variables? Instead of putting qualitative variables (1, 2, or 3) it would be good to put quantitative variables such as total distance of the trail, duration of the route… Please clarify…

Line 215: Maybe change “habits” by “biological types”.

Line 228: I don’t understand why the number of species (252) it’s different with line 213 (384 sp.). It’s should be the same number… Please clarify.

Figure 4: Are there significant difference between routes?? For that, the authors need more data (more plots).

Line 267: I still do not understand the relevance of this to the objective of the work.

Table 2 (line 280): Change the name of the month “Set” by “Sep”.

Line 287: The authors should indicate number of species in the rainy seasons and dry season. For example (number of species in rainy season= Average and standard deviation; number of species in dry season= Average and standard deviation).

Lines 294 and 297: Would is the p-value in Pearson correlations?

Line 297: The p-value better with two decimals (R=0.15)

Line 327: As there are several types of vegetation, plots should be made in the different types. The number of plots should be increased.

Lines 389 - 404: I think this section has nothing to do with the objective of the work. The values of the level of biodiversity (diversity of native plants) are already obtained with alpha and beta and not with their phenological state.

Line 412: I am still not clear what are the ecosystem and cultural criteria. Because I don't see anything in the results analysed about archaeological complexes.

Line 427: If it is a Protected Natural Area, housing construction should not be allowed.

Line 434: Preparing and implementing the replacement of eucalyptus plantations is restoration.

Author Response

Response to Revisor 2:

On the sampling methodology of plant diversity

In principle we consider the sampling effort of data collection is sufficient, a) two types of sampling were made to try to cover the total diversity and richness. Of these, the qualitative one was based on registering all the species of vascular plants existing in the routes, the quantitative data to make the calculations of diversity, alpha and gamma, through variables of coverage, abundance.

The distances on each route are long, however we have taken as a reference the main types of vegetation, beginning, middle and end.

Another fundamentation is that the diversity of plants in this work is considered as one of the components of the integral approach together with the cultural one, for use in the ecotourism activity.

Some comments in the document:

Line 97: What vegetation type in this region? Please, indicate the community’s vegetation (Phytosociology).

Was clarified

Line 100: The authors comment that two well-marked periods. Please, the authors should indicate the average annual temperature and annual rainfall in this two periods.

Figure 1: The author should include scale and orientation in the two small maps on the right side of figure.

Was implemented

Line 11: Are there any statistical surveys that indicate that these places are the most visited sites? On what basis? On direct observation?

R: As indicated in the document, these routes are the ones that are traditionally the ones that are frequently traveled for tourism activities. Also, part of these statements are dealt with in the discussion section in L 423, 424.

Line 137: Change “20 x 50 mm” by “20 x 50m”. And it’s better change “0.1 hectare” by “1000 m2”.

Was clarified

Line 137: On such a long distance (e.g. 10-12 km of a route) the number of plots is insufficient. Only 3 plots you can't conclude anything concrete as you have too few sampling points. What is the distance between plots? Please clarify…

Was clarified

Line 139: I understand that “crown area” is cover (%), it’s correct? Please clarify…

Was implemented

Line 139: Numbers of individual included all individuals (juveniles and mature)?

Was implemented

Line 169: Other analysed that I recommend to the author is a DCA

(detrended correspondence analysis) with CANOCO or R software.

We believe that the analyses carried out are sufficient.

Line 176:  I do not understand what the flowering period has to do with the objective of the work, which is to identify the potential of ten traditional nature trails to be used for ecotourism activities in the inter-Andean valley of Cusco. The authors are not answering the question with this section.

It was clarified and implemented in introduction, methodology and discussions

Line 185: How did you obtain data on precipitation and average temperatures? Have you placed dataloggers on the plots to obtain this information? Please clarify…

Was clarified

Lines 201-203: With respect to number of units of cultural remains, these refer to the presence of archaeological remains corresponding mainly to archaeological complexes including temples, ruins, rooms and others belonging to the pre-Inca or Inca cultures, the authors do not indicate where they have obtained this information. Please clarify…

Indicated and clarified in L: 210 – 212

Was implemented

Line 203: How has the anthropogenic impact of each route been studied? What variables have the authors used? Have they not taken into account quantitative variables? Instead of putting qualitative variables (1, 2, or 3) it would be good to put quantitative variables such as total distance of the trail, duration of the route… Please clarify…

R: We consider qualitative data because they are the most approximate and accessible for measurement. We do not have quantitative data because for this we need more measurement time and resources, which we do not have. Nor is the objective of the work the exact calculation of the anthropic effects.

Line 215: Maybe change “habits” by “biological types”.

Was implemented

Line 228: I don’t understand why the number of species (252) it’s different with line 213 (384 sp.). It’s should be the same number… Please clarify.

Was clarified

Figure 4: Are there significant difference between routes?? For that, the authors need more data (more plots).

Was explained

Line 267: I still do not understand the relevance of this to the objective of the work.

Was explained

Table 2 (line 280): Change the name of the month “Set” by “Sep”.

Was implemented

Line 287: The authors should indicate number of species in the rainy seasons and dry season. For example (number of species in rainy season= Average and standard deviation; number of species in dry season= Average and standard deviation).

It was implemented

Lines 294 and 297: Would is the p-value in Pearson correlations?

Was clarified

Line 297: The p-value better with two decimals (R=0.15)

Was clarified

Line 327: As there are several types of vegetation, plots should be made in the different types. The number of plots should be increased.

Was explained

Lines 389 - 404: I think this section has nothing to do with the objective of the work. The values of the level of biodiversity (diversity of native plants) are already obtained with alpha and beta and not with their phenological state.

Was implemented

Line 412: I am still not clear what are the ecosystem and cultural criteria. Because I don't see anything in the results analysed about archaeological complexes.

Was implemented

Line 427: If it is a Protected Natural Area, housing construction should not be allowed.

Was clarified

Line 434: Preparing and implementing the replacement of eucalyptus plantations is restoration.

Round 2

Reviewer 2 Report

I have noticed that the work has been improved and the proposed comments have been added by the authors, which I believe has enriched the research work.

Best regards